# Maternal behavior influences vocal practice and learning processes in the greater sac-winged bat

**Ahana Aurora Fernandez[1,2]\*, Nora Serve[2], Sarah-Cecil Fabian[2], Mirjam Knörnschild[1,3,4]**

[1]Museum für Naturkunde, Leibniz Institute for Evolution and Biodiversity Science, Berlin, Germany; [2]Department of Biology, Chemistry, and Pharmacy, Institute of Biology, Free University, Berlin, Germany; [3]Smithsonian Tropical Research Institute, Balboa, Panama; [4]Evolutionary Ethology, Institute for Biology, Humboldt-Universität zu Berlin, Berlin, Germany

**\*For correspondence:**
fernandez.aurora.ahana@gmail.com

**Competing interest:** The authors declare that no competing interests exist.

## eLife Assessment

This **important** study provides insights into the role of maternal behavior in the learning and ontogeny of vocalization. It finds evidence that the maternal behavior of sac-winged bats (Saccopteryx bilineata) can influence the learned territorial songs of their pups. The behavioral analyses are **convincing**, using longitudinal acoustic recordings and behavioral monitoring of individual mother-pup pairs across development and multiple wild bat colonies. The work will be relevant to a broad audience interested in the evolution and development of social behavior as well as sensory-motor learning.

**Abstract** Learning, particularly vocal learning, is often a social process. In human infants, it is well-established that social interactions influence speech acquisition and are hypothesized to modulate attentiveness and sensory processes, thereby affecting the speech-learning process. However, our understanding of how social interactions shape vocal ontogenetic processes in non-human mammals, particularly those which vocally learn, remains limited. In the bat *Saccopteryx bilineata*, pups acquire the adult vocal repertoire through a distinctive babbling behavior that shows interesting similarities to human infant babbling. While babbling encompasses many different syllable types, it is particularly noteworthy that pups learn song syllables by imitating adult singing males. The pups' social environment involves frequent interactions with their mothers, whereas adult males mainly serve as the primary source of acoustic input. We monitored the vocal ontogeny of wild pups, investigating whether their social environment influenced three aspects of babbling: the amount of vocal practice, the pups' final syllable repertoire size and the production of the syllable types acquired through vocal learning. The results demonstrate that maternal behavioral displays significantly influence the amount of vocal practice, the presence and versatility of song syllable types in babbling and the percentage of mature song syllables. Our findings show that maternal feedback plays a significant role in the vocal ontogeny and learning processes of *S. bilineata*, thus enhancing our understanding of the relationship between social feedback and vocal development in mammalian vocal learners.

## Introduction

Vocal development and learning are often fundamentally social processes (*Carouso-Peck et al., 2021*; *Doupe and Kuhl, 1999*; *García, 2019*; *Gultekin and Hage, 2017*; *Janik and Slater, 1997*; *Takahashi et al., 2015*). Vocal production learning (VPL) is a continuous trait encompassing various

**eLife digest** Babies learn to speak by imitating the sounds made by those around them. This process is shaped not only by their passive exposure to language, but also by the smiles, gestures and other forms of encouragement displayed by their caregivers.

The impact of this social feedback has been extensively researched in people, yet it is still poorly understood in the rare non-human species that also learn their vocalizations by imitating adults. Most studies in such animals have focused on interactions between a single tutor and tutee in captivity, failing to acknowledge the influence of other individuals in the wild.

Found in the rainforests of Central and South America, greater sac-winged bats provide a unique opportunity to explore the social mechanisms of vocal learning. Much like human babies, their pups babble; they spent nearly a third of their daytime activity vocalizing and practising the songs they hear from adult males. However, it is with their mothers that these pups most frequently interact.

To investigate whether maternal feedback influences vocal learning, Fernandez et al. followed nineteen greater sac-winged pups from wild colonies in Panama and Costa Rica over two breeding seasons. They focused on three vocal learning measures: how long pups practiced babbling for, the number and diversity of song syllables they produced, and how 'mature' these were compared to those of adult males. The team also tracked how often mothers interacted with their respective babies (by hovering overhead or touching them, for example), as well as the number of adult males in the vicinity.

Statistical analyses showed that pups with more involved mothers babbled longer each day, and for more days overall; they also produced a wider variety of adult-like sounds, and a higher percentage of syllables that sounded like those produced by an adult singer. In contrast, the number of singing males nearby did not improve any measure of vocal learning. By shedding light on how social feedback influences song acquisition in non-human species, the findings by Fernandez et al. help to better understand the factors that shape vocal learning, and the evolution of language.

---

levels, including vocal convergence (i.e. gradual change of vocalizations based on social input) and vocal imitation of species-specific or heterospecific vocalizations and artificial sounds (*Janik and Knörnschild, 2021*). The social environment influences all levels of VPL to varying degrees, either passively through acoustic input or actively through vocal and behavioral feedback. For example, the social environment of goat kids passively influences the acoustic convergence of their innate contact call (*Briefer and McElligott, 2012*). Songbird species require acoustic input for successful song learning but vary in the degree to which they require social interactions (*García, 2019*). Social feedback has also been shown to influence the maturation patterns of the adult vocal repertoire in marmosets (*Gultekin and Hage, 2017*). To gain a deeper understanding of vocal learning and development, it is important to explore the social factors that influence it, the extent of their impact, and the consequences when these factors are absent. Questions about how the social environment influences vocal development and learning have begun to receive more research attention (*Bistere et al., 2024*; *Carouso-Peck and Goldstein, 2019*; *Chen et al., 2016*; *Gultekin and Hage, 2017*; *Takahashi et al., 2016*; *Takahashi et al., 2017*; *West and King, 1988*). However, there is need for more studies that explore interactions beyond the tutor-tutee relationship. Only a few studies have addressed this to date (i.e. zebra finch: *Bistere et al., 2024*; *Carouso-Peck and Goldstein, 2019*; i.e. brown-headed cowbird: *West and King, 1988*), demonstrating the significance of this research direction. Further studies are needed – particularly in the wild (*García, 2019*) – to broaden the range of study species, with a special focus on mammalian vocal learners. In this study, we explored the role of social feedback in one of the few non-human mammalian vocal learners (*Janik and Knörnschild, 2021*), the greater sac-winged bat *Saccopteryx bilineata*. This neotropical bat species is known for its extensive vocal repertoire, which includes male song (*Behr and von Helversen, 2004*). During ontogeny, pups acquire a part of the adult vocal repertoire by imitating the territorial song of adult male tutors (*Knörnschild et al., 2010*). This learning process is expressed through a vocal practice known as 'pup babbling' (*Fernandez et al., 2021*; *Knörnschild et al., 2006*). Pup babbling shows parallels to human infant babbling (*Fernandez et al., 2021*), which makes *S. bilineata* a particularly exciting mammalian vocal-learning species to study. Babbling is a conspicuous and unique behavior:

Over the course of 7 weeks, pups spend almost 30% of their active daytime hours babbling. Babbling bouts are 7 min on average but can last up to 43 min (*Fernandez et al., 2021*). Babbling bouts are composed of multisyllabic repetitive vocal sequences (i.e. syllable trains) containing adult-like syllable types (i.e. syllable types of the adult vocal repertoire) and undifferentiated proto-syllables (i.e. syllables only produced by babbling pups) (*Fernandez et al., 2021*). Both male and female pups babble (*Fernandez et al., 2021*), which is different from the male-biased plastic song production found in songbirds (*Doupe and Kuhl, 1999*; *Marler, 1970*) (but see *Odom et al., 2014* for female bird song). Although babbling bouts can result in maternal care in the form of nursing (*Strauss et al., 2010*), babbling is clearly not a begging behavior; babbling differs from begging in structural composition (i.e. including adult and juvenile syllables) and its lack of context specificity (*Ter Haar et al., 2021*). To solicit maternal care pups specifically produce isolation calls which they deliver in a fast repetitive manner. Pup babbling is not tied to a specific function – pups either babble alone in the day-roost or interact with their mothers, mainly through various additional behaviors which can occur singly or in interactive sequences (*Figure 1A–C*). Remarkably, females never produce these behavioral displays outside of the babbling context (*Bradbury and Emmons, 1974*; *Tannenbaum, 1975*; *Voigt et al., 2008*). Adult males do not behaviorally interact with pups, but their vocalizations provide acoustic input in the form of conspecific adult song (*Behr and von Helversen, 2004*).

We studied the vocal ontogeny of 19 pups from 6 wild colonies during two consecutive field seasons (May-September 2016–2017) in Panama and Costa Rica (see Materials and methods). In the present study, we investigated how two primary social factors—the number of singing males (i.e. tutors) and the maternal behaviors displayed during a babbling bout—influenced the following three aspects of pup vocal development and learning: (1) the amount of vocal practice, (2) final syllable repertoire size and, (3) various characteristics of the song syllables learned through vocal imitation while babbling.

To assess maternal influence, we counted the total number of maternal behavioral displays occurring during babbling, focusing only on interactive behaviors and excluding comfort behaviors (*Appendix 1—table 1*). Territorial song is produced by adult territorial males daily at dusk and dawn for acoustic territorial defence (*Behr and von Helversen, 2004*). Thus, the number of singing males (i.e. tutors) present in a colony was used as a proxy for the amount of social acoustic input a given pup received during its ontogeny. Pup age was included in analyses to account for physical maturation.

## Results

### Amount of vocal practice

The amount of vocal practice was defined as (a) the length of daily practice, that is babbling bout duration in seconds, and (b) the overall babbling phase duration, that is time in days from the first day of babbling until the last (see Materials and methods). The total number of maternal behaviors performed during babbling bouts, as well as pup age, had a significant effect on the duration of daily babbling bouts, whereas the number of tutors had no effect (*Table 1*, *Figure 1D*). This finding does not result from the fact that longer bouts simply offer the possibility for more maternal displays (see *Appendix 1—table 2*). Interestingly, some mothers were more active than others in the sense that they showed more and different behavioral displays when interacting with pups. Therefore, we calculated a maternal activity score for each female to capture its overall behavioral activity (see Materials and methods). We found that the amount of maternal activity significantly influenced the duration of the babbling phase, whereas, once again, the number of male tutors did not (*Table 1*, *Figure 1E*). These findings show that maternal behaviors significantly shape the amount of time pups spend vocally practicing—age-related increases in endurance and physical maturity explain only part of the variance (*Table 1*).

### Final repertoire size

The final repertoire size is defined as the number of adult-like syllable types (i.e. syllables that were identifiable as syllables of the later adult vocal repertoire) present in pup babbling at the time point of weaning (analyzed for a prior study *Fernandez et al., 2021*, which reports the relevant methods).

The maternal activity score was not significantly correlated with the pups' final syllable repertoire size (Spearman rank correlation: $r_s$ = 0.05, p=0.88, N=10 mother-pup dyads). Likewise, the number of singing males in a day-roost did not correlate with the pups' final syllable repertoire size (Spearman

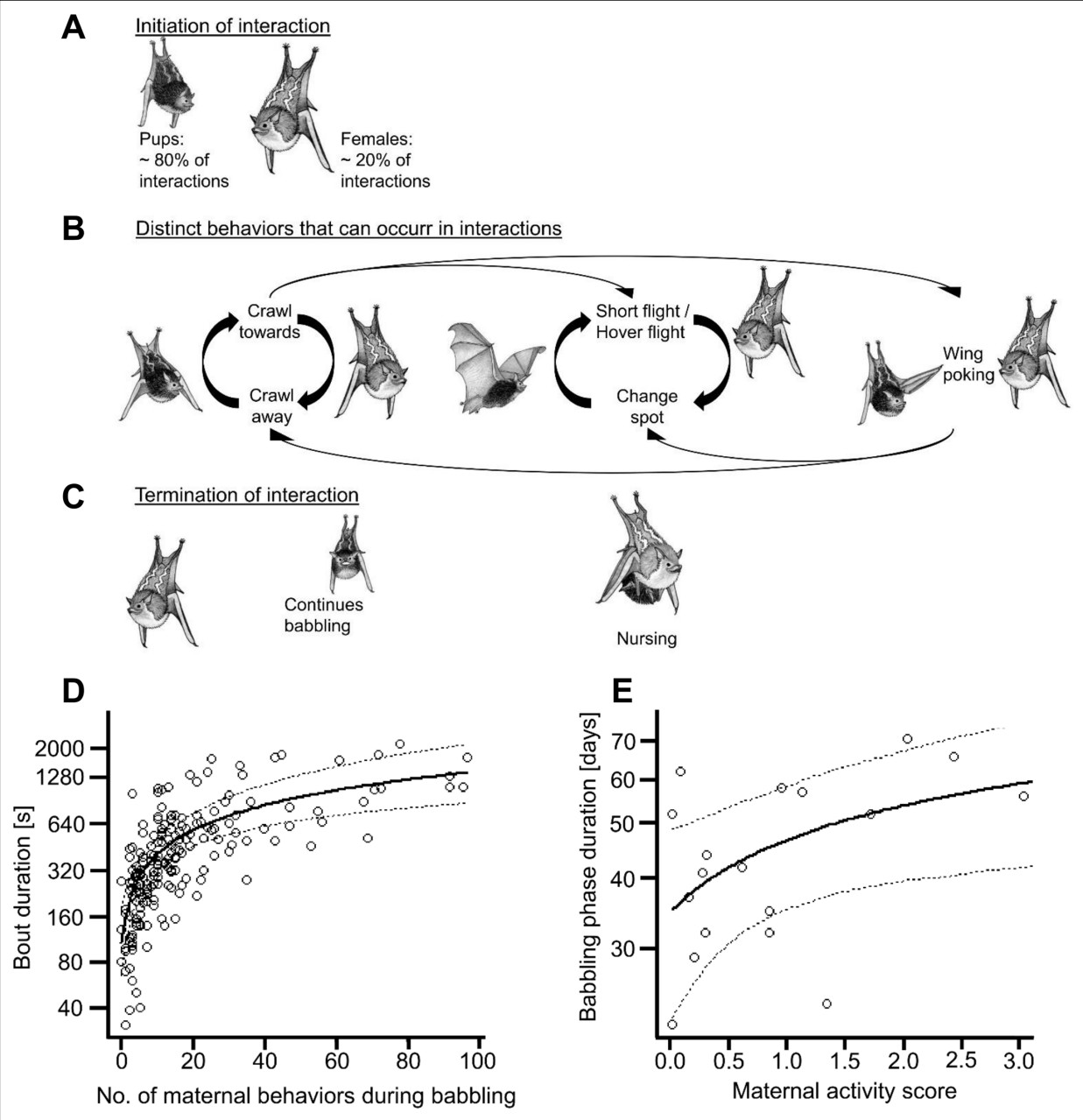

**Figure 1.** Behavioral displays and interactions during babbling and maternal influence on the amount of vocal practice. (**A-C**) shows the typical social behaviors and interactions of mother-pup pairs during a babbling bout. Note: all these behaviors could occur singly or in an interactive way (i.e. a behavioral display was followed by another in less than one second). (**A**) In on average 80% of cases, the interaction was initiated by the pup, normally by crawling toward the mother while babbling. A mother initiated interactions by hovering in front of or landing next to her pup. (**B**) After the start of an interaction, mother and pup engaged in a number of different behaviors, which were performed equally by both mother and pup (*Appendix 1—table 1*), with the exception of 'wing poking', which was mostly observed in pups (*Appendix 1—table 1*). The different behaviors could variably succeed after one another (large arrows); however, 'crawl towards' and 'crawl away', as well as 'short flights' respectively 'hover flights' and a temporarily 'change in the roosting spot' were often observed in repetitive sequences (bold black arrows). Hover flights in front of the interactive partner were the most conspicuous behavior, followed by the short flights either next to the interaction partner or within the day roost. With increasing pup age, short flights became progressively longer, for example pups even briefly left the day roost and landed on an adjacent tree. (**C**) Interactions between mother and pup could be terminated in two ways: Either the pup continued to babble alone or the mother allowed nursing. Maternal behavior did not differ with regard to pup sex (SI). Drawing credit: C. A. S. Mumm. (**D**) shows the positive influence of the maternal behaviors displayed during babbling bouts on babbling bout duration [s] (N=19 pups, N=186 babbling bouts, GLMM, family Gamma with log link). (**E**) shows that higher maternal activity scores led to a longer babbling phase duration [days] (N=19 pups, LMER). The figure legends represent the raw data, whereas the fitted lines (with lower (2.5th percentile) and upper (97.5th percentile) confidence intervals) are based on the calculated model. Neither bout duration nor babbling phase duration differed between pup sex (SI).

**Table 1.** The influence of the social environment on the amount of vocal practice.
Response variables (first column): A: *Babbling bout duration [s]:* the babbling bout duration measured in seconds (N=19 pups, N=186 babbling bouts, GLMM, family Gamma with log link, random factor pup ID: repeated measurements over time of the same focal pups). B: *Babbling phase duration:* The number of days pups spent babbling (N=19 pups, LMER, random factor colony ID: measurements of pups from the same colony). The predictor variables for both models were z-transformed, a standardization procedure that facilitates convergence of the model. The second column depicts the fixed and random effects for both models. The third column depicts the estimate and the standard error as well as the standard deviation for the random effect, and the last column the p-value (bold if influence is significant). Abbreviations: 'Mat. behav.'=maternal behaviors.

| Response variable | Parameter | Estimate (SE) | Significance |
|---|---|---|---|
| | *Fixed effects* | | |
| | Intercept | 6.038 (0.074) | p<0.001 |
| | Mat. behav. during babbling (standardized) | 0.537 (0.044) | **p<0.001** |
| | Pup age (standardized) | 0.100 (0.044) | **p=0.022** |
| | No. of tutors (standardized) | –0.041 (0.077) | p=0.6 |
| | *Random effect (standard deviation)* | | |
| A *Babbling bout duration [s]* | Pup ID | 0.19 | |
| | *Fixed effects* | | |
| | Intercept | 44.322 (5.222) | p=0.001 |
| | Maternal activity score (standardized) | 7.317 (2.824) | **p=0.02** |
| | No. of tutors (standardized) | 3.560 (5.167) | p=0.530 |
| | *Random effect (standard deviation)* | | |
| B *Babbling phase duration [days]* | Colony ID | 11.20 | |

rank correlation: $r_s$ = 0.52, p=0.13, N=10 pups). However, we found a trend for a positive correlation between the mean babbling bout duration of pups and their final syllable repertoire size (Spearman rank correlation: $r_s$ = 0.58, p=0.08, N=10 pups).

## Learned song syllables

Babbling *S. bilineata* pups use vocal imitation to acquire the syllables of their territorial song (***Knörn-schild et al., 2010***). Babbling is therefore a behavioral indicator of learning. During babbling, the pup's song precursors strongly resemble the songs of their acoustic tutors, regardless of their genetic relatedness to the tutor. The acoustic changes which the song syllables undergo cannot solely be attributed to maturation processes or convergence toward a species mean, indicating that pups are influenced by the songs they hear on a daily basis (***Knörnschild et al., 2010***). To understand how the social environment impacts learning processes, we examined the characteristic 'buzz' syllable types of the territorial song (***Figure 2A***). The territorial song is composed of five different song syllable types (***Figure 2—figure supplement 1***). However, pups differ in which and how many of these five different song syllable types are present in their babbling. We therefore calculated for each babbling bout the song syllable type versatility (see Materials and methods): high values indicate that the pups' vocal practice contained many of these different types, low values indicate that the pups' vocal practice contained fewer different syllable types. Furthermore, the most common song syllable type (***Appendix 1—table 3***) was categorized into mature and precursor syllables (see Materials and methods) to assess whether the social environment had an influence on the presence of mature syllables in babbling. Subsequently, we investigated whether the social environment influenced the (a) total number of song syllables present in a babbling bout, (b) the song syllable type versatility in a babbling bout, and (c) the percentage of mature versus precursor syllables of the most common song syllable type in babbling bouts. Both maternal behaviors and pup age positively influenced the total number of buzz syllables in babbling bouts, while the number of male tutors had no effect (***Table 2***, ***Figure 2B***). Similarly, the versatility was influenced by maternal behavior and pup age, with no influence from the number of male tutors (***Table 2***, ***Figure 2C***). The maternal behavior

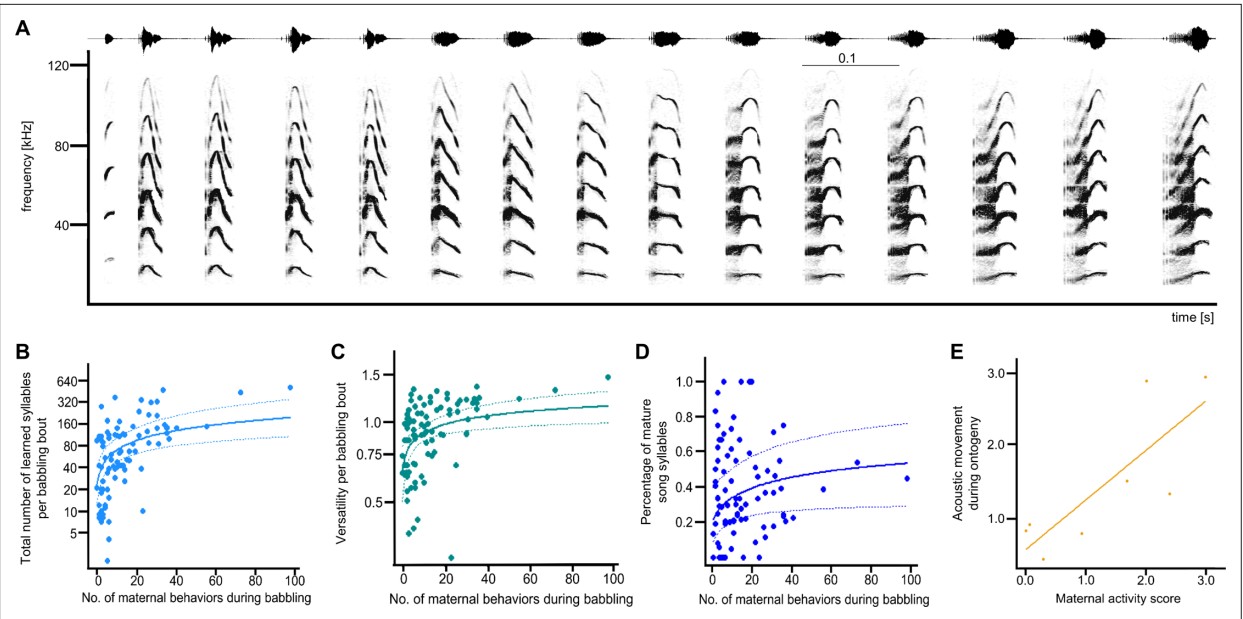

**Figure 2.** Maternal influence on learned syllables in babbling bouts. (**A**) shows a typical multisyllabic territorial song produced by an adult male. The most characteristic syllable type is called 'B2' (i.e. the last six syllables of this song), composed of a buzzed part connected to a tonal part. This is also the most common song syllable type in pup babbling (***Appendix 1—table 3***). The spectrogram was created in Avisoft SasLab Pro (Hamming window, 1024 FFT, 50% overlap resulting in 317 Hz frequency and 2.048ms time resolution). (**B**) shows the positive significant influence of maternal behavioral displays on the total number of learned syllables within babbling bouts (N=10 pups, N=90 babbling bouts, GLMM, family negative binomial with log link). Territorial songs can be flexibly composed of different buzz syllable types (***Figure 2—figure supplement 1***). (**C**) shows that maternal behavior positively influences the versatility (i.e. number of different learned syllable types, ***Figure 2—figure supplement 2***) of babbling bouts (N=10 pups, N=90 babbling bouts, LMER). (**D**) The most common song syllable type B2 (***Appendix 1—table 3***) was categorized into precursor and mature syllables (***Figure 2—figure supplement 3***). Maternal behavioral displays positively influenced the percentage of mature song syllables (***Table 3***) (N=10 pups, N=85 babbling bouts, GLMM family binomial with logit link). The figure legends represent the raw data, whereas the fitted lines (with lower (2.5th percentile) and upper (97.5th percentile) confidence intervals) are based on the calculated model. (**E**) shows the acoustic movement of pups during their vocal ontogeny based on acoustic parameters of the syllable type B2, the most common song syllable type (N=8 pups, N=220 syllable trains, ***Figure 2—figure supplement 4***). The larger the value on the y-axis, the larger the acoustic movement during the pup's vocal ontogeny (***Figure 2—figure supplement 5***). The acoustic movement was positively correlated with the maternal activity score.

The online version of this article includes the following figure supplement(s) for figure 2:

**Figure supplement 1.** Five different song syllable types in babbling bouts.

**Figure supplement 2.** Babbling excerpt with syllable labeling.

**Figure supplement 3.** Precursor versus mature syllables.

**Figure supplement 4.** B2 syllable measurements for acoustic change tracking during ontogeny.

**Figure supplement 5.** Acoustic movement during ontogeny.

also influenced the percentage of mature song syllables present in babbling, with no effect of pup age and the number of male tutors (***Table 2***, ***Figure 2D***). Additionally, we quantified the acoustic change of the most common learned syllable type (B2, ***Appendix 1—table 3***) during ontogeny and explored its relationship with maternal activity. We used classical acoustic parameters (e.g. duration, peak frequency) and linear frequency cepstral coefficients (LFCCs) to assess both traditional acoustic features and timbre (voice color) of B2 syllables (see Materials and methods). Discriminant function analyses were performed to obtain centroids (mean canonical scores for each individual) for the early and late babbling phase (see Materials and methods). Subsequently, we calculated Euclidean distances between the early and late babbling phase for each individual, where larger values indicated greater acoustic changes during the babbling phase. Our findings revealed a positive correlation between acoustic changes and maternal activity (Spearman rank correlation: $r_s$ = 0.73, p=0.0037, ***Figure 2E***).

**Table 2.** The influence of the social environment on learned syllables.

Response variables: A: *Total number of song syllables*: the learned territorial song syllables which were present in a babbling bout, (N=10 pups, N=90 babbling bouts, GLMM, family negative binomial with log link, random factor ID: repeated measurements over time of the same focal pups). B: *Song syllable versatility*: how many of the five different song syllable types (*Figure 2—figure supplement 1*) were present in a babbling bout, (N=10 pups, N=90 babbling bouts, LMER, random factor ID: repeated measurements over time of the same focal pups). Neither the total number of song syllables nor song syllable versatility differed between pup sex (SI). C: *Percentage of mature song syllables*: We investigated the influence of different predictor variables on the percentage of mature song syllables of the most common syllable type, B2 (*Appendix 1—table 3* and N=10 pup, N=85 babbling bouts, GLMM family binomial with logit link, random factor ID: repeated measurements over time of the same focal pups, random factor observation-level: to avoid overdispersion). The predictor variables for both models were z-transformed, a standardization procedure that facilitates convergence of the model. Fixed and random effects are depicted in the second column. The third column depicts the estimate with standard error, and the last column the p-value (significant influence indicated in bold). Abbreviation: 'Mat. behav.'=maternal behaviors.

| Response variable | Parameter | Estimate (SE) | Significance |
|---|---|---|---|
| | *Fixed effects* | | |
| | Intercept | 4.188 (0.194) | p<0.001 |
| | Maternal behavior during babbling (standardized) | 0.460 (0.098) | **p<0.001** |
| | Pup age (standardized) | 0.446 (0.096) | **p<0.001** |
| | No. of tutors (standardized) | 0.036 (0.193) | p=0.9 |
| | *Random effects (standard deviation)* | | |
| A *Total number of song syllables* | Pup ID | 0.55 | |
| | *Fixed effects* | | |
| | Intercept | 0.889 (0.030) | p<0.001 |
| | Maternal behavior during babbling (standardized) | 0.105 (0.031) | **p<0.001** |
| | Pup age (standardized) | 0.125 (0.030) | **p<0.001** |
| | No. of tutors (standardized) | −0.022 (0.030) | p=0.479 |
| | *Random effects (standard deviation)* | | |
| B *Song syllable versatility* | Pup ID | 0.04 | |
| | *Fixed effects* | | |
| | Intercept | −0.698 (0.355) | p=0.049 |
| | Maternal behavior during babbling (standardized) | 0.376 (0.188) | **p=0.045** |
| | Pup age (standardized) | 0.185 (0.177) | p=0.296 |
| | No. of tutors (standardized) | 0.107 (0.341) | p=0.754 |
| | *Random effects (standard deviation)* | | |
| | Pup ID | 0.98 | |
| C *Percentage of mature song syllables* | Observation-level | 1.24 | |

## Discussion

Our findings demonstrate that both maturational effects (i.e. pup age) and social feedback (i.e. maternal feedback) together influenced various aspects of pup babbling behavior, whereas the presence of male tutors had no influence on any of these aspects. In *S. bilineata*, babbling very likely serves multiple functions which are not mutually exclusive. One function of babbling is the acquisition of the learned syllables of the territorial song. The territorial song is an important acoustic signal for adult bats (*Behr et al., 2009*; *Knörnschild et al., 2017*). In *S. bilineata*, unlike most other mammals, females disperse after weaning, while most males stay in their natal colony (*Nagy et al., 2007*). Adult males queue for territory access. Once they obtain a territory, they mark it through territorial singing,

**Table 3.** Terminology.

| Term | Description |
|---|---|
| Syllable type | Sound unit with characteristic spectro-temporal features |
| Undifferentiated proto-syllables | Syllables only produced by pups, occurring in babbling and vanish after weaning. |
| Adult-like syllable | Syllables which are present in the later adult vocal repertoire (after weaning). |
| Buzz syllable types of song | Territorial song syllables with discernible pulsed part; 5 distinct syllable types: B1, B2, B2 trill, B3, B4 (only syllable with smeared buzz part) (*Figure 2—figure supplement 1*). |
| Mature syllable | Mature syllables show no gap (e.g. between the pulsed and tonal part, within the buzzed part) (*Figure 2—figure supplement 3*). |
| Precursor syllable | Distinct gaps within the pulsed part and/or between pulsed and tonal part (*Figure 2—figure supplement 3*). |
| Early babbling phase | First half of entire babbling phase |
| Late babbling phase | Second half of entire babbling phase |
| Song syllables | Syllables of which the territorial song is composed of (*Figure 2A*) |

a daily activity influenced by various social factors (*Eckenweber and Knörnschild, 2013*). The territorial song encodes a group signature (*Eckenweber and Knörnschild, 2013*) that may function as a password to identify social group members and potentially assists in the cooperative defense of the colony against unrelated intruders (*Nagy et al., 2012*). This group signature might be acquired during the babbling phase, when pups listen to singing adult males. An intriguing aspect of this species is that, unlike most song-learning bird species, female pups show no differences from males in babbling behavior and vocal development (*Fernandez et al., 2021*; *Knörnschild et al., 2010*). This study corroborated this finding: female pups received the same maternal feedback, and their song syllable imitation did not differ in any way from male pups (as observed as well in *Knörnschild et al., 2010*). This phenomenon is rare among vocal learners and raises the question of why female pups match male vocal development despite not using the learned vocalizations later in life. One potential explanation might lie in the function of the territorial song for adult females: it serves as an acoustic signal to help females locate new suitable colonies after dispersal. The territorial song exhibits different dialects, with females showing a preference for local over foreign dialects (*Knörnschild et al., 2017*). The own early practice and production of song might enhance the ability to evaluate male song and support mating decisions. The female preference for the dialect learned during ontogeny likely drives the selection of precise imitative learning in males. This might explain the exceptional and long vocal learning and practice phase during ontogeny. It could also explain why maternal feedback during this period positively influences the production (i.e. amount and versatility) of learned syllables. The process of vocal imitation involves learning through self-perception: An individual hears its own vocal output and matches it to the acoustic input of the tutor (*Brainard and Doupe, 2000*). Increased practice of learned syllable types could strengthen the connection between acoustic perception and vocal production in a manner akin to the mechanisms observed in human infants and songbirds (*Doupe and Kuhl, 1999*; *Kuhl, 2007*). Furthermore, such practice is likely to improve fine-scale control of the vocal apparatus in a way that will eventually allow for the production of a mature syllable or song. Our data show that *S. bilineata* mothers also influence the presence of mature song syllables (*Table 3*) of the most commonly produced syllable type within babbling bouts. In human infants, contingent social feedback, both vocal and non-vocal (e.g. touch, smiling, moving closer to the infant), can positively influence babbling (*Goldstein et al., 2003*; *Goldstein and Schwade, 2008*). For example, social feedback can increase the rate of production of mature speech sounds in babbling (*Goldstein et al., 2003*). However, it should be noted that this finding does not indicate that contingent social feedback increases the size of the infant's babbling repertoire through imitation, but rather that it induces a shift toward the production of mature speech sounds that are already present in the infant's repertoire. The debate about how much imitation is present in babbling during social interactions, and how this imitation manifests itself, is still ongoing. One interesting possibility is that infants imitate prosodic contours in dyadic interactions with their caregivers (*Gratier and Devouche, 2011*). In most songbird studies, similarity between tutor and tutee defines song learning success (*García, 2019*). While this study did not assess the similarity between pups and their adult tutors (due to the limited availability

of song data from all adult tutors), we were able to investigate the acoustic trajectories of individual pups during their babbling phase. Our data demonstrates that higher maternal activity was associated with larger overall changes in acoustic parameters in the most commonly learned syllable type (B2). This finding serves as an initial indication that non-vocal interactions with the mother may influence a pup's individual learning trajectories. Future studies will focus on the relationship between acoustic change, maternal feedback, and learning success, specifically investigating contingencies between particular pup vocalizations and maternal displays in natural settings. Playback experiments are an additional approach to test the impact of contingency on vocal learning. For example, one study in zebra finches demonstrated that contingent non-vocal maternal feedback affects imitation success (*Carouso-Peck and Goldstein, 2019*), while another recent study found that female calls can promote song learning, but the role of contingency remains to be determined (*Bistere et al., 2024*). Beyond the territorial song, it remains unclear whether and which other syllables from the adult vocal repertoire are acquired through vocal learning. However, producing a diverse vocal repertoire that includes syllables with distinct spectro-temporal features may necessitate practice, regardless of whether they are learned socially or not. Vocal practice also involves the transitions between the different syllable types, and possibly even the acquisition of specific sequential orders for later multisyllabic adult vocalizations.

While we did not find a direct correlation between maternal feedback and the final syllable repertoire size of pups, babbling time may play a role in expanding the vocal repertoire, as indicated by the positive correlation trend between the final syllable repertoire size and the mean bout duration. Maternal behaviors significantly increased the amount of vocal practice (*Figure 1D–E*, *Table 1*), which might indirectly support the maturation and acquisition of a large vocal repertoire. Such a relationship has already been documented in common marmosets, where social feedback influences the maturation (i.e. the development of the adult calls) of the vocal repertoire (*Gultekin and Hage, 2017*). It may appear surprising that the number of singing males did not influence any aspects of pup vocal practice and learning. In songbirds, acoustic input from and social interaction with the tutor plays a pivotal role in song learning (*Chen et al., 2016*; *García, 2019*). In *S. bilineata*, direct interactions between pups and adult males were absent. Nevertheless, it is possible that the amount of adult male singing could influence pup practice. The present study cannot directly address this relationship, since the measure of song activity applied—the number of singing males—is an indirect indicator. In future studies, we will include the total number and duration of songs to which pups are exposed to and examine whether this might influence various aspects of their vocal practice and learning. Furthermore, the findings of this study are purely observational and future captive playback studies will provide further insight into how social effects affect vocal practice and learning.

In this study, we focused on how social feedback shaped babbling. However, babbling—itself a salient vocal signal—likely also influences social feedback (*Ter Haar et al., 2021*). This might also apply to *S. bilineata* where pups initiated ~80% of social interactions, suggesting that maternal feedback is likely influenced by the pup's vocal practice. In humans, parents are sensitive to their infants' vocalizations, can intuitively assess speech maturation, and are more likely to respond to sounds that resemble speech (*Abney et al., 2017*; *Albert et al., 2018*; *Oller et al., 2001*). This can initiate a positive feedback loop: when infants receive contingent adult feedback for their speech-like utterances, they tend to produce more speech-like vocalizations (*Goldstein and Schwade, 2008*; *Warlaumont et al., 2014*). Babbling in humans is also hypothesized to serve as a vocal fitness signal, potentially evolved to elicit more parental care thus, gaining importance throughout human evolution (*Oller et al., 2016*). In this context, it would be highly interesting to study the role of hormones and neuromodulators (*Theofanopoulou et al., 2017*). In the case of *S. bilineata*, it would be intriguing to investigate whether babbling elicits oxytocin release in mothers and whether, in turn, this hormone modulates maternal social feedback.

In conclusion, our study shows that maternal social feedback shapes different aspects of the vocal practice and learning of pups. As bats are highly social mammals that share key aspects of human brain architecture and vocal development, our findings will substantially advance our knowledge about how social interactions shape vocal learning.

# Materials and methods

**Key resources table**

| Reagent type (species) or resource | Designation | Source or reference | Identifiers | Additional information |
|---|---|---|---|---|
| Biological sample (*Saccopteryx bilineata*) | *Saccopteryx bilineata*; *S. bilineata*; greater sac-winged bat | other | N/A | Wild animals: two different populations (Costa Rica and Panamá), both male and female pups (N=19 mother-pup pairs) 'see Material and methods, sections Study species, study sites, acoustic and behavioral data collection' |
| Software, algorithm | SPSS | IBM SPSS Statistics | RRID:SCR_002865 | |
| Software, algorithm | MATLAB | MATLAB | RRID:SCR_001622 | |
| Software, algorithm | R | R Studio | RRID:SCR_000432 | |
| Software, algorithm | Avisoft SASLab-Pro | Avisoft Bioacoustics | RRID:SCR_014438 | |
| Software, algorithm | VLC | VLC media player | N/A | |

## Study species

*Saccopteryx bilineata* roosts in perennial stable groups, that is colonies (*Tannenbaum, 1975*). Females synchronize the birth of a single pup around the beginning of the rainy season (i.e. May). Pups are weaned at 10–12 weeks of age (*Tannenbaum, 1975*). Day-roosts of *S. bilineata* are located in tree cavities or outer walls of man-made structures (*Yancey et al., 1998*). *Saccopteryx bilineata* is very light-tolerant and most social interactions (e.g. courtship, territorial defense, babbling behavior) occur during the day within the roost (*Behr and von Helversen, 2004*; *Fernandez et al., 2021*; *Knörnschild et al., 2006*). Individuals in the day-roost maintain an inter-individual distance of 5–8 cm (*Voigt et al., 2008*). Only pups are allowed to ignore this inter-individual distance without eliciting aggression. This species possesses a large vocal repertoire (i.e. 25 distinct syllable types) which is fully delineated (*Behr and von Helversen, 2004*; *Fernandez et al., 2021*; *Knörnschild and von Helversen, 2008*). These syllable types are composed into mono- or multisyllabic vocalizations. For most vocalizations, the behavioral context and function is known as well (e.g. the territorial song is an aggressive signal used during territorial defense).

## Study sites

Acoustic recordings and observations of social behaviors were recorded ad libitum sensu (*Altmann, 1974*) throughout the pups' ontogeny (i.e. from birth until weaning) in their day-roost. In 2016, we conducted acoustic recordings and additionally monitored the accompanying behaviors from ten mother-pup dyads belonging to three colonies in the natural reserve Curú in Costa Rica. In 2017, we likewise conducted acoustic recordings and simultaneous behavioral observations from nine mother-pup dyads belonging to three colonies in Gamboa, a field station of the Smithsonian Tropical Research Institute in Panama.

## Acoustic and behavioral data collection

In 2016 and 2017, acoustic recordings and behavioral observations of mother-pup dyads during babbling were performed simultaneously. During all field seasons, we counted the number of singing males present in a colony, representing the indirect social environment of our focal pups. Since adult bats are individually banded with colored plastic rings on their forearms (A.C. Hughes Ltd. UK, size XCL, one band per forearm) and mothers only nurse their own, single pup, individual identity of our focal bats could be determined. The inter-individual distance outside of the mother-pup pairs and discrimination of the individually banded mothers between own and alien pups allowed for focal acoustic recordings and behavioral monitoring. After habituation, it was possible to record and observe focal bats within a distance of 2–4 m without noticeable disturbance.

During both field seasons, focal pups were recorded at least twice per week, during alternating morning or afternoon recording sessions (approximately 5-hr total recording time) throughout their vocal ontogeny (i.e. from birth until weaning, for more details see methods in *Fernandez et al., 2021*). We used high-quality ultrasonic recording equipment (500 kHz sampling rate, 16-bit depth

resolution) to record the vocalizations. The recording set-up consisted of a microphone (Avisoft Ultra SoundGate 116Hm, with condenser microphone CM16, frequency range 1–200 kHz±3 dB, connected to a laptop computer Lenovo S21e) running the software Avisoft-Recorder (v4.2.05 R. Specht, Avisoft Bioacoustics, Glienicke, Germany). The microphone was mounted on a tripod, which was positioned in close distance to the day-roost and directed at the respective focal pup. Behaviors of mother-pup dyads during babbling were monitored using a combination of video recordings and simultaneously taking notes, to ensure that all behaviors were captured. The video camera was mounted next to the microphone on the tripod.

## Acoustic data analysis

Acoustic analyses of babbling were performed on two different levels: the syllable level (i.e. the basic unit) and the babbling bout level (i.e. composed of syllable trains and interspersed with silent intervals; see Materials and methods *Fernandez et al., 2021*).

The amount of vocal practice was measured in two ways: the duration (in seconds) of babbling bouts and the duration of the babbling phase. A babbling bout is composed of at least three syllable trains which belong to at least two different syllable train categories and has a minimum duration of 30 s (for details see *Fernandez et al., 2021*). A syllable train is defined as a sequence of at least five syllables (for details see *Fernandez et al., 2021*). A babbling bout was either terminated if a pup remained silent for longer than three minutes and/or when a pup was allowed to nurse. The babbling phase duration (in days) spans the period during which the pups were observed to engage in babbling. The day of birth was used to calculate pup age in general, age at babbling onset and the duration of the babbling phase (for details see *Fernandez et al., 2021*). The daily bout duration (in seconds) and babbling phase duration (in days) was obtained during both field seasons (N=186 babbling bouts in total, Costa Rica 2016: N=123 bouts from 10 pups, Panama 2017: N=63 bouts from 9 pups). The final repertoire size of 10 pups (Costa Rica 2016) was analyzed for a previous study (*Fernandez et al., 2021*; method details can be found there). The number of adult-like syllable types (i.e. the syllables that were identifiable as syllables of the later adult vocal repertoire) present in pup babbling at the time point of weaning represented the pups' final syllable repertoire size. The territorial song is composed of different syllable types whose presence varies with behavioral context (*Eckenweber and Knörnschild, 2013*; *Knörnschild et al., 2017*; *Knörnschild et al., 2016*). The typical territorial song (*Figure 2A*) which is sung at dawn and dusk starts with simple, short tonal syllables and ends with composite syllables (with pulsed and tonal components) which are called buzz syllables. The tonal and buzz syllables are categorized into different syllable types (*Fernandez et al., 2021*). For this study, we focused on the buzz syllable types because those are the syllables imitated from tutors (*Knörnschild et al., 2010*) and they encode most information about the signaler (i.e. individual-, group and population signature) (*Eckenweber and Knörnschild, 2013*; *Knörnschild et al., 2017*; *Knörnschild et al., 2016*). Moreover, the simple tonal syllable types are also present in other vocalization types, some of which are present directly after pup birth. To analyze the influence of the mothers' behavior on the buzz syllable types we selected babbling bouts of 10 pups (N=90 babbling bouts, Costa Rica 2016) where we had simultaneous behavioral observations and acoustic recordings (with good signal-to-noise ratio). If possible, we selected one babbling bout per week and pup during the babbling phase and manually labeled all present buzz syllables (*Figure 2—figure supplement 2*). We defined five different buzz syllable types based on their spectro-temporal properties, namely B1, B2, B2 trill, B3, and B4 (*Figure 2—figure supplement 1*) following our categorization in *Fernandez et al., 2021*. Our visual classification was statistically confirmed in an earlier study (*Fernandez et al., 2021*), and syllables in this study were classified by the same observer. In total, we manually labeled 9201 buzz syllables (N=90 babbling bouts, Costa Rica 2016). We then performed the following measures: (a) count of all present buzz syllables (but without B1 because it was not present in each pup) and (b) the buzz syllable versatility (R: *vegan* package, function diversity) including all five syllable types present in a babbling bout. Furthermore, we identified the number of mature and precursor syllables of the most common syllable type B2 (*Appendix 1—table 3* and N=4847 B2 syllables, N=90 babbling bouts, Costa Rica 2016). Syllables were classified as precursor if there were distinct gaps within the pulsed part and/ or within the pulsed and tonal part (*Figure 2—figure supplement 3*). This is clearly not the case in mature syllables of adult males (*Figure 2A*, *Figure 2—figure supplement 3*). We focused on B2 syllables because this was the most abundant syllable type in babbling bouts for all pups and

because it is the syllable type characteristic of the typical adult male song (*Figure 2A*). To investigate the influence of the social environment on the change of acoustic properties during ontogeny, we measured 530 B2 syllables from eight pups (for two pups of the Costa Rica dataset there was not enough data from the entire ontogeny; therefore, they were excluded for this analysis) and 205 B2 syllables from adult males (for explanation see statistical analysis below). Again, we focused on B2 syllables because this was the most abundant syllable type in babbling bouts for all pups and because it is the syllable type characteristic of the typical adult male song (Figur 2 A). To be able to investigate the acoustic trajectory during ontogeny, we measured B2 syllables from different babbling bouts throughout the entire vocal ontogeny of each pup. For each pup, we normally selected three B2 syllables from one to three syllable trains per babbling bout (N=220 trains in total). It was not always possible to measure three B2 syllables from three trains in a babbling bout either because there were simply not so many B2 syllables (especially at the beginning of the babbling phase) or because the signal-to-noise ratio quality was not sufficient for acoustic measurements. To quantify the acoustic change during ontogeny, we separated the entire babbling period into an early and late babbling phase (for more details see statistical analysis below). For each pup, we analysed the same number of syllable trains per babbling phase (7–20 trains per pup for each babbling phase). Subsequently, we measured various acoustic parameters of the B2 syllables. For our acoustic parameter measurement analysis, we used the software Avisoft SASLab Pro (v.5.3.01, R. Specht, Avisoft Bioacoustics, Glienicke, Germany). Before measurements, we converted the sampling frequency of each acoustic recording from 500kHz to 250kHz to obtain a higher frequency resolution. The buzz syllables are composed of a pulsed (buzz-like) part preceding a tonal part. We extracted the same acoustic parameters for the pulsed and tonal part separately (*Figure 2—figure supplement 4*). Acoustic parameters were subsequently averaged (separately for pulsed and tonal parts) for each train to minimize temporal dependence among syllables produced within close succession. Spectrograms for measuring syllable parts were created using a Hamming window with 1024–point Fourier transformation and 93.75% overlap (frequency resolution: 244 Hz, time resolution: 0.256ms). Syllable parts were multi-harmonic; measurements were taken from the harmonic containing most energy (i.e. clearest spectral shape) and the extracted acoustic parameters were subsequently calculated to the fundamental frequency (first harmonic) to make measurements comparable across the different syllable parts. For each syllable part, we measured one temporal (duration) and six spectrum-based parameters (peak-, minimum-, maximum frequency, bandwidth, entropy and harmonics-to-noise ratio). These spectrum-based parameters were measured at the center of each syllable part and at 9 points evenly distributed over each syllable (start, end, and seven intermediate locations). In addition, the six spectrum-based parameters were averaged over each syllable part (mean values) and the maximum spectral values of the entire syllable part were calculated. The spectrum-based parameters were used to estimate the frequency curvatures and entropy and harmonics-to-noise curvatures (henceforth referred to as entropy curvatures) of the syllable parts. Furthermore, we extracted acoustic features that were based on linear-frequency cepstral coefficients (LFCCs) since these capture important acoustic characteristics of bat vocalizations (*Fernandez and Knörnschild, 2020*; *Knörnschild et al., 2017*). Each LFCC describes the spectral properties of an entire acoustic signal, comprising its most important features in a compact form. The extracted features summarize both common acoustic parameters (e.g. peak frequency) and timbre (voice color). We used a customized MATLAB script in the toolbox 'voicebox' (v. R2014a) for the feature extraction. Each vocalization sequence was composed of B2 syllables of the same train containing the first four harmonics (F0-F3, frame length 0.01 s). We extracted 20 LFCCs from each vocal sequence and used them for subsequent statistical analyses.

## Behavioral data analysis

To describe and quantify the behaviors of mothers and pups during babbling bouts, videos were analyzed using VLC (VLC media player), complemented by field notes. We established an ethogram (*Appendix 1—table 1*) based on our observations of mother-pup behaviors and the behavioral observations of a former study (*Strauss et al., 2010*). Inter-observer-reliability was 100% among the observers. Based on this ethogram, we counted the maternal behavioral displays (N=186 babbling bouts, N=3222 interactive behaviors, N=19 mothers) occurring during babbling, focusing only on the interactive behaviors, excluding comfort behaviors (see *Appendix 1—table 1*).

To assess the mothers' influence over the course of the entire pup babbling phase, we calculated a maternal activity score (based on 186 babbling bouts). We calculated the behavioral quotient for each babbling bout (i.e. number of maternal interactive behaviors divided by the bout duration) and subsequently averaged these quotients for each female. We calculated also the mean hover rate per bout. We counted hover behaviors (i.e. hovering in mid-air in front of a roosting conspecific) separately because it is a very distinctive behavior that is never performed outside of a social context; for instance, hovering is produced by adult males during courtship in a multimodal display (*Behr and von Helversen, 2004*). The mean behavioral quotient and the mean hover rate were summed up, and we obtained a maternal activity score ranging from 0.03 to 3.03. Lower numbers accounted for less active mothers and higher numbers for more active ones. The activity score and the number of analyzed bouts did not correlate significantly (i.e. the activity score of a mother is not biased by the different number of bouts analyzed per mother-pup pair; Spearman-rank correlation test: $r_s = 0.27$, p=0.25).

## Statistical analysis

All statistical analyses were performed in R (RStudio 2021, version 1.4.1106) and SPSS (IBM SPSS Statistics 2022, version 29.0.0.0.(241)). Statistical differences were considered significant for p<0.05, *p<0.05, **p<0.01, ***p<0.001. Since we did not obtain enough high-quality recordings and/or behavioral observations for all mother-pup dyads, we had to restrict some analyses to a data subset. Therefore, sample sizes differed for our analyses. All generalized linear mixed models (GLMMs) and LMERs were fitted using the package *lme4* and *lmerTest* in R. In all multivariate models, the predictor variables were z-transformed (standardized: i.e. meaning the substraction of the mean and division by the standard deviation), a standardization procedure that facilitates convergence of the model (*Gelman and Hill, 2006*). For standardization we used the function *scale()*. In all models, the variable maternal behavior was log-transformed before it was standardized. This transformation was applied to reduce the influence of individual high values, which can have a disproportionate effect on the model. The (negative) binomial models were checked for overdispersion using the package *blemco*. Overdispersion has only to be checked for in distributions which do not have variance parameters (e.g. poisson or binomial models).

## The amount of vocal practice

To assess the influence of maternal behaviors on bout duration we counted all interactive maternal behaviors (*Appendix 1—table 1*) during a babbling bout. We analyzed a total of 3222 maternal behaviors produced by mothers during 186 babbling bouts (Costa Rica 2016: N=123 babbling bouts, ten mother-pup dyads, Panama 2017: N=63 babbling bouts, nine mother-pup dyads). To assess the maternal influence on bout duration we calculated a GLMM (Gamma family, with log link), with bout duration as response variable, maternal behaviors (log transformed and subsequently standardized), pup age (standardized) and the number of tutors (standardized) as fixed factors and pup ID as random factor (based on the datasets of 2016 and 2017, N=19 mother-pup dyads, N=186 babbling bouts). To assess the influence of the social environment on babbling phase duration (in days), we calculated a LMER. An LMER was chosen because it fits the distribution of the raw data better than a GLMM. A GLMM with Gamma-family would not adequately represent such a symmetrical distribution. The response variable was babbling phase duration (in days). The fixed effects were the maternal activity score (standardized) and the number of tutors (standardized; N=19 pups, N=186 babbling bouts), and colony ID was included as random factor. To investigate potential sex differences in bout duration, babbling phase duration and maternal interactive behaviors we calculated Mann-Whitney-U tests (N=131 babbling bouts, N=14 mothers; using the r-package 'coin'). Not all pups were sexed; therefore, we calculated sex differences only for 14 out of the 19 pups.

## The pups' final repertoire size

We calculated three separate Spearman rank correlation tests (*cor.test* function, method 'spearman') to investigate the relationship between the female activity score, respectively the number of tutors, the mean bout duration and the pups` final repertoire size (N=10 mother-pup dyads from 2016, repertoire size see *Fernandez et al., 2021*).

## The properties of the learned buzz syllables

To assess the maternal influence on the number of buzz syllables in a babbling bout, we calculated a GLMM (negative binomial model) with number of buzz syllables as response variable (without syllable type B1 because it was not produced by each pup), maternal behaviors (log transformed and subsequently standardized), pup age (standardized) and the number of tutors (standardized) as fixed factors and pup ID as random factor (based on the dataset of 2016, N=10 mother-pup dyads, N=90 babbling bouts, N=8837 buzz syllables). A negative binomial model was chosen over a poisson model because it better fitted the data structure. To assess the influence of the social environment on the buzz syllable versatility within a babbling bout, we calculated the Shannon-wiener Index (*vegan* package) for each babbling bout (N=10 mother-pup dyads, N=90 babbling bouts, N=9201 buzz syllables). This index was the response variable, whereas the maternal behaviors (log transformed and subsequently standardized), pup age (standardized), and the number of tutors (standardized) were the fixed factors with pup ID as random factor (lmer model). An LMER was chosen because a Gamma-family cannot deal with the value 0. Furthermore, we calculated the effect of the social environment on the presence of mature song syllable types (focusing on the most common syllable type B2, *Appendix 1—table 3*) in babbling bouts. We used the function *cbind(Mature syllables, precursor syllables)* to analyze the effect of the predictor variables on the presence of mature syllables. The maternal behavior (log transformed and subsequently standardized), the pup age (standardized) and the number of tutors (standardized) were the overdispersion as random factors (GLMM, binomial, N=85 babbling bouts, N=4847 buzz syllables).

## The pups' acoustic movement during ontogeny

For each syllable part of the B2 syllable (i.e. buzz part and tonal part) we separately obtained derived parameters (i.e. frequency and entropy curvatures) by performing four separate principal component analyses (PCA) with varimax rotation on the parameters mentioned above (see acoustic data analysis) which reduced multicollinearity between original acoustic parameters. Furthermore, we decided to calculate separate PCAs for the colonies that were acoustically separated (i.e. the bats could not hear each other). PCA frequency curvature: 36 frequency parameters, that is peak, maximum, and minimum frequency, and bandwidth at 9 points evenly distributed over each syllable part; PCA entropy curvature: 18 entropy and harmonics-to-noise ratio parameters, that is entropy and harmonics-to-noise ratio at 9 points evenly distributed over each syllable part. All PCAs fulfilled Kaiser-Meyer-Olkin (KMO) and Bartlett's test criteria, ensuring the appropriateness of our data for PCAs.

Subsequently, we conducted two separate discriminant function analyses (DFAs) for the two colonies, respectively. The DFAs were adjusted to the unequal number of analyzed syllable parts per pup and male by computing group sizes based on prior probabilities. We split the dataset into two phases, namely an early and late babbling phase (i.e. babbling phase duration divided by two). We calculated the multidimensional signal space with the acoustic data from the early babbling phase (entire babbling phase divided by two, first half: early phase, second half: late phase, *Figure 2—figure supplement 5*) and from the adult males and subsequently plotted the acoustic data from the late babbling phase into this signal space. This approach is like a test and training approach where a multidimensional space is calculated based on the training data and subsequently, the test data is analyzed based on the training set. We chose to integrate syllables from adult males to generate the signal space (DFA 1=early babbling phase), since it is known that pups become more similar to their tutors during ontogeny (*Knörnschild et al., 2010*). Instead of including different adult males we summarized the syllables as a representative 'generic' male, as we had unequal (and sometimes insufficient) number of syllables from individual males. For each pup, we had the same number of syllable trains per babbling phase (range: 7–20 trains per babbling phase, N=8 pups). For both DFAs, we included the same parameters, namely, duration of the buzz part, duration of the tonal part, mean peak, minimum and maximum frequency of the buzzed respectively tonal part, and two derived parameters of the buzzed respectively tonal part, describing entropy curvatures (all with eigenvalues >1). Furthermore, we included the first two linear frequency cepstral coefficients (LFCC1, LFCC2). In total, we included 12 variables in both separate DFAs.

To assess the acoustic movement pups underwent during their babbling phase, we calculated the Euclidean distances between individual centroids of babbling phase 1 and phase 2 based on the DFAs. For each pup, we calculated the distance between itself in babbling phase 1 and babbling phase 2 (range distances: 0.46–2.96, with higher numbers depicting larger acoustic movement during ontogeny). Subsequently, we correlated the individual pup movements with the female activity index.

## Acknowledgements

We thank the Smithsonian Tropical Research Institute and the Natural Reserve Curú for excellent research conditions. We thank S Ripperger for valuable comments on the manuscript. This work was supported by grants from the Elsa-Neumann Foundation to AAF and from the European Research Council under the European Union's Horizon 2020 Programme (2014–2020)/ERC GA 804352 to MK.

## Additional information

### Funding

| Funder | Grant reference number | Author |
| --- | --- | --- |
| Elsa-Neumann Stipendium des Landes Berlin | | Ahana Aurora Fernandez |
| European Research Council | 10.3030/804352 | Mirjam Knörnschild |

The funders had no role in study design, data collection and interpretation, or the decision to submit the work for publication.

### Author contributions

Ahana Aurora Fernandez, Conceptualization, Data curation, Formal analysis, Funding acquisition, Investigation, Visualization, Methodology, Writing – original draft; Nora Serve, Investigation; Sarah-Cecil Fabian, Formal analysis; Mirjam Knörnschild, Supervision, Funding acquisition, Visualization, Project administration, Writing – review and editing

### Author ORCIDs

Ahana Aurora Fernandez  https://orcid.org/0000-0002-4298-4736
Mirjam Knörnschild  https://orcid.org/0000-0003-0448-9600

### Ethics

All data collection in Panama (2017) was approved by the local governmental authorities and the Smithsonian Tropical Research Institute Animal Care and Use Committee (ACUC 2017-0516-2020). All data collection in Costa Rica (2016) was approved by the Costa Rican Ministry of the Environment (SINAC-SE-CUS-PI-R-088-2016).

Reviewer #1 (Public review): https://doi.org/10.7554/eLife.99474.3.sa1
Author response https://doi.org/10.7554/eLife.99474.3.sa2

## Additional files

### Supplementary files

MDAR checklist

### Data availability

Data can be found on Dryad: https://doi.org/10.5061/dryad.7wm37pw4s.

The following dataset was generated:

| Author(s) | Year | Dataset title | Dataset URL | Database and Identifier |
|---|---|---|---|---|
| Fernandez AA, Serve N, Fabian SC, Knörnschild M | 2025 | Data from: Maternal behavior influences vocal practice and learning processes in the greater sac-winged bat | https://doi.org/10.5061/dryad.7wm37pw4s | Dryad Digital Repository, 10.5061/dryad.7wm37pw4s |

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

# Appendix 1

**Appendix 1—table 1.** Ethogram of behaviors during babbling bouts.

| Behavior | Description | Observed in: |
|---|---|---|
| crawl away* | One individual crawls a certain distance (>2 cm) away from the other. | P / M |
| crawl towards* | One individual crawls towards another individual (approaches up to 2 cm). | P / M |
| hover* | Hover flight in front of an individual (for a few seconds). | P / M |
| wing-poke* | Poking a conspecific with wrist. Pups normally repeatedly poke their mothers. Mothers usually poke the pup only once to terminate the babbling bout. Includes wing pokes with and without physical contact. | P / M |
| short flight* | Individual flies to another spot in the day roost or individual flies towards interaction partner. | P / M |
| rock | Rocking with entire body from side to side; also used when mother wants pup to detach from teat. | P / M |
| wing stretch | Wing is stretched out completely. | P / M |
| head stretch | Head and neck are conspicuously bent and stretched to either left or right side. | P |
| push up | Push wrists against wall and push body up. | P / M |
| dribble | Pound with wrists against wall, very fast and repetitive. | P |
| wrist lift | Pound with wrists against wall once, not hitting other individual. | P |
| babble | Vocal practice. | P |

The behaviors labeled with a * either occurred as solitary behavior or as behavioral sequence (i.e. interaction). These interactive behaviors were used to assess the maternal behavioral influence on pup babbling. The behaviors without a * (with the exception of babble) were defined as comfort behaviors and not included as behavioral feedback. The third column depicts if the behavior was observed in the pup (P) or both pup and mother (P/M).

**Appendix 1—table 2.** Maternal display rates: duration of babbling is not a confounding factor.

| Response variable | Parameter | Estimate (SE) | Significance |
|---|---|---|---|
| | *Fixed effects* | | |
| | Intercept | 6.132 (0.091) | p<0.001 |
| | Mat. behav. rates during babbling (standardized) | –0.146 (0.053) | **p=0.006** |
| | Pup age (standardized) | 0.418 (0.050) | **p<0.001** |
| | No. of tutors (standardized) | –0.063 (0.095) | p=0.51 |
| | *Random effect (standard deviation)* | | |
| **A** *Babbling bout duration [s]* | Pup ID | 0.24 | |

The number of displays in longer bouts could just reflect that more displays are possible in a longer period. To assess this potential confounding factor, we investigated the effect of display rates on vocal bout duration (N=19 pups, N=186 babbling bouts, GLMM family Gamma with log link).

## Maternal influence on the amount of vocal practice

### Maternal behaviors
The number of analyzed bouts did not bias the analyzed maternal behavioral response (Spearman-rank correlation test: p=0.8, rho = –0.07, N=19 pups). The number of maternal behaviors did not differ between pup sex (Mann-Whitney-U-Test: z=–0.063888, p=1, N=14 pups).

### Bout duration
Bout duration did not differ between pup sex (Mann-Whitney-U-Test: z=–0.44721, p=0.7, N=14 pups) or between colonies (Kruskal-Wallis Test: chi-squared=7.6853, p=0.2, N=19 pups). However, bout duration was different between the two study populations, Costa Rica and Panama, with longer mean bout durations found in Panama (Mann-Whitney-U-Test: z=–2.0412, p=0.04, N=19 pups, average bout durations: Costa Rica:~7 min, Panama:~10 min).

### Babbling phase duration
Babbling phase duration did not differ between pup sex (Mann-Whitney-U-Test: z=–0.51279, p=0.6, N=14 pups) or between colonies (Kruskal-Wallis Test: chi-squared, 11,224, p=0.047, N=19 pups, followed by pairwise Wilcox. test with Bonferroni correction: ns). However, the babbling phase duration was different between the two populations, with longer babbling phases present in pups from Costa Rica (Mann-Whitney-U-Test: z=2.5754, p=0.009, N=19 pups, average babbling phase duration: Costa Rica: 52.7 days, Panama: 36.3 days).

## The influence of the social environment on learned syllables

### Total number of syllables within babbling bouts
Female and male pups did not differ in the number of buzz syllables they produced during babbling (Mann-Whitney-U-Test: z=2.1213, p=0.06, N=7 pups). Likewise, there was no significant difference in the amount of buzz syllables produced between the different colonies (Kruskal-Wallis-Test: chi-squared=2.0218, p=0.37, N=10 pups).

### Syllable versatility
There was no difference between the syllabic diversity between male and female pups (Mann-Whitney-U-Test: z=–1.4142, p=0.23, N=7 pups) or between colonies (Kruskal-Wallis-Test: chi-squared=0.54909, p=0.76, N=10 pups).

**Appendix 1—table 3.** The average proportion of syllable type production.

| ID | B1 | B2 | B2-trill | B3 | B4 |
|----|-----|------|----------|------|------|
| 11 | 0 | 47.8 | 6.6 | 24.7 | 12.6 |
| 12 | 25.8 | 56.1 | 6.6 | 9.7 | 1.8 |
| 13 | 0.13 | 50.2 | 12.7 | 16.1 | 6.7 |
| 14 | 0 | 54.9 | 25.8 | 27.8 | 19.2 |
| 15 | 0.1 | 54.9 | 16.2 | 9.0 | 19.7 |
| 16 | 4.9 | 67.2 | 1.6 | 20.6 | 5.7 |
| 17 | 2.3 | 73.5 | 3.4 | 20.4 | 0.4 |
| 18 | 0 | 54.2 | 14.5 | 24.1 | 7.2 |
| 19 | 0 | 53.7 | 7.8 | 24.8 | 13.8 |
| 20 | 0 | 50.9 | 24.5 | 11.6 | 5.3 |

For each pup, the average proportion of each learned syllable type produced during all analysed babbling bouts is displayed. Averaged over all pups B2 is the most commonly produced syllable type within babbling bouts (53.8%), followed by B3 (18.6%), B2 trill (11.4%), B4 (8.9%), and B1 (2.8%).

