## [Editor Report · eLife Assessment]

This **important** study provides insights into the role of maternal behavior in the learning and ontogeny of vocalization. It finds evidence that the maternal behavior of sac-winged bats (Saccopteryx bilineata) can influence the learned territorial songs of their pups. The behavioral analyses are **convincing**, using longitudinal acoustic recordings and behavioral monitoring of individual mother-pup pairs across development and multiple wild bat colonies. The work will be relevant to a broad audience interested in the evolution and development of social behavior as well as sensory-motor learning.

---

## [Referee Report · Reviewer #1 (Public review)]

Summary:

Fernandez et al. investigate the influence of maternal behavior on bat pup vocal development in Saccopteryx bilineata, a species known to exhibit vocal production learning. The authors performed detailed longitudinal observations of wild mother-pup interactions to ask whether non-vocal maternal displays during juvenile vocal practice, or 'babbling', affect vocal production. Specifically, the study examines the durations of pup babbling events and the developmental babbling phase, in relation to female display rates, as well as pup age and the number of nearby singing adult males. Furthermore, the authors examine pup vocal repertoire size and maturation in relation to maternal display rates encountered during babbling. Statistical models identify female display behavior as a predictor of (i) babbling bout duration, (ii) the length of the babbling phase, (iii) song composition and (iv) syllable maturation. Notably, these outcomes were not influenced by the number of nearby adult males (the pups' source of song models) and were largely independent of general maturation (pup age). These findings highlight the impact of non-vocal aspects of social interactions in guiding mammalian vocal development.

Strengths:

Historically, work on developmental vocal learning has focused on how juvenile vocalizations are influenced by the sounds produced by nearby adults (often males). In contrast, this study takes the novel approach of examining juvenile vocal ontogeny in relation to non-vocal maternal behavior, in one of the few mammals known to exhibit vocal production learning. The authors collected an impressive dataset from multiple wild bat colonies in two Central American countries. This includes longitudinal acoustic recordings and behavioral monitoring of individual mother-pup pairs, across development.

The identified relationships between maternal behavior and bat pup vocalizations have intriguing implications for understanding the mechanisms that enable vocal production learning in mammals, including human speech acquisition. As such, these findings are likely be relevant to a broad audience interested in the evolution and development of social behavior as well as sensory-motor learning.

Weaknesses:

The authors qualitatively describe specific patterns of female displays during pup babbling, however, subsequent quantitative analyses are based on aggregate measures of female behavior that pool across display types. Consequently, it remains unclear how certain maternal behaviors might differentially influence pup vocalizations (e.g. through specific feedback contingencies or more general modulation of pup behavioral states).

Comments on revisions:

(1) More detailed analyses of female behavior may be beyond the scope of this study, given the nature of the dataset/recordings. I look forward to the authors' future work on this aspect.

By addressing the important distinction between display number vs. display rate, the authors have provided more direct support for the claim that babbling behavior is related to female displays.

(2) The additional information regarding exposure to adult male song is appreciated.

(3) Added discussion of pup sex differences provides useful context and intriguing speculation about the role of female pup babbling.

(4) The authors' additions have significantly improved the clarity of their acoustic terminology and syllable analyses.

---

## [Author Response]

The following is the authors’ response to the original reviews

**Public Reviews:**

**Reviewer #1 (Public review):**
Summary:Fernandez et al. investigate the influence of maternal behavior on bat pup vocal development in Saccopteryx bilineata, a species known to exhibit vocal production learning. The authors performed detailed longitudinal observations of wild mother-pup interactions to ask whether non-vocal maternal displays during juvenile vocal practice or 'babbling', affect vocal production. Specifically, the study examines the durations of pup babbling events and the developmental babbling phase, in relation to the amount of female display behavior, as well as pup age and the number of nearby singing adult males. Furthermore, the authors examine pup vocal repertoire size and maturation in relation to the number of maternal displays encountered during babbling. Statistical models identify female display behavior as a predictor of (i) babbling bout duration, (ii) the length of the babbling phase, (iii) song composition, and (iv) syllable maturation. Notably, these outcomes were not influenced by the number of nearby adult males (the pups' source of song models) and were largely independent of general maturation (pup age). These findings highlight the impact of non-vocal aspects of social interactions in guiding mammalian vocal development.

We thank Reviewer 1 for the time and effort dedicated to the revision of our study. The suggestions for the revision of our manuscript were very helpful and have improved our manuscript considerably.

Strengths:Historically, work on developmental vocal learning has focused on how juvenile vocalizations are influenced by the sounds produced by nearby adults (often males). In contrast, this study takes the novel approach of examining juvenile vocal ontogeny in relation to non-vocal maternal behavior, in one of the few mammals known to exhibit vocal production learning. The authors collected an impressive dataset from multiple wild bat colonies in two Central American countries. This includes longitudinal acoustic recordings and behavioral monitoring of individual mother-pup pairs, across development.The identified relationships between maternal behavior and bat pup vocalizations have intriguing implications for understanding the mechanisms that enable vocal production learning in mammals, including human speech acquisition. As such, these findings are likely to be relevant to a broad audience interested in the evolution and development of social behavior as well as sensory-motor learning.

We thank reviewer 1 for this assessment.

Weaknesses:The authors qualitatively describe specific patterns of female displays during pup babbling, however, subsequent quantitative analyses are based on two aggregate measures of female behavior that pool across display types. Consequently, it remains unclear how certain maternal behaviors might differentially influence pup vocalizations (e.g. through specific feedback contingencies or more general modulation of pup behavioral states).In analyzing the effects of maternal behavior on song maturation, the authors focus on the most common syllable type produced across pups. This approach is justified based on the syllable variability within and across individuals, however, additional quantification and visual presentation of categorized syllable data would improve clarity and potentially strengthen resulting claims.

We agree that our analysis of maternal behaviour does not investigate potential contingencies between particular maternal behavioural displays and pup vocalizations (e.g. particular syllable types). Our data collected for this study on maternal behaviour includes direct observations, field notes and/or video recordings. In the future, it will be necessary to work with high-speed cameras for the analysis of potential contingencies between particular maternal behavioural displays and specific pup vocalizations, which allow this kind of fine-detailed analysis. We have planned future studies investigating whether pup vocalizations elicit contingent maternal responses or vice versa. In the revision of our manuscript, we have included a comment pointing out that this special behaviour will be investigated in greater detail in the future.

As suggested by reviewer 1, in our revised manuscript we have included more information on methods to improve understandability. In particular, we have:

-presented more information on different steps of our acoustic analyses

-provided additional and clearer spectrogram figures representing the different syllable types and categorizations

-changed the figures accompanying our GLMM analyses following the suggestion of Reviewer 1

**Reviewer #2 (Public review):**
Summary:This study explores how maternal behaviors influence vocal learning in the greater sac-winged bat (Saccopteryx bilineata). Over two field seasons, researchers tracked 19 bat pups from six wild colonies, examining vocal development aspects such as vocal practice duration, syllable repertoire size, and song syllable acquisition. The findings show that maternal behaviors significantly impact the length of daily babbling sessions and the overall babbling phase, while the presence of adult male tutors does not.The researchers conducted detailed acoustic analyses, categorizing syllables and evaluating the variety and presence of learned song syllables. They discovered that maternal interactions enhance both the number and diversity of learned syllables and the production of mature syllables in the pups' vocalizations. A notable correlation was found between the extent of acoustic changes in the most common learned syllable type and maternal activity, highlighting the key role of maternal feedback in shaping pups' vocal development.In summary, this study emphasizes the crucial role of maternal social feedback in the vocal development of S. bilineata. Maternal behaviors not only increase vocal practice but also aid in acquiring and refining a complex vocal repertoire. These insights enhance our understanding of social interactions in mammalian vocal learning and draw interesting parallels between bat and human vocal development.

We thank reviewer 2 for his/her time and effort dedicated to the revision of our study. The suggestions were very helpful in improving our manuscript.

Strengths:This paper makes significant contributions to the field of vocal learning by looking at the role of maternal behaviors in shaping the vocal learning phenotype of Saccopteryx bilineata. The paper uses a longitudinal approach, tracking the vocal ontogeny of bat pups from birth to weaning across six colonies and two field seasons, allowing the authors to assess how maternal interactions influence various aspects of vocal practice and learning, providing strong empirical evidence for the critical role of social feedback in non-human mammalian vocal learners. This kind of evidence highlights the complexity of the vocal learning phenotype and shows that it goes beyond the right auditory experience and having the right circuitry.The paper offers a nuanced understanding of how specific maternal behaviors impact the acquisition and refinement of the vocal repertoire, while showing the number of male tutors - the source of adult song - did not have much of an effect. The correlation between maternal activity and acoustic changes in learned syllable types is a novel finding that underscores the importance of non-vocal social interactions in vocal learning. In vocal learning research, with some notable exceptions, experience is often understood as auditory experience. This paper highlights how, even though that is one important piece of the puzzle, other kinds of experience directly affect the development of vocal behavior. This is of particular importance in the case of a mammalian species such as Saccopteryx bilineata, as this kind of result is perhaps more often associated with avian species.Moreover, the study's findings have broader implications for our understanding of vocal learning across species. By drawing parallels between bat and human vocal development (and in some ways to bird vocal development), the paper highlights common mechanisms that may underlie vocal practice and learning in both humans and other mammals. This interdisciplinary perspective enriches the field and encourages further comparative studies, ultimately advancing our knowledge of the evolutionary and developmental processes that shape vocal productive learning in all its dimensions.

We thank reviewer 2 for this assessment.

Weaknesses:Some weaknesses can be pointed out, but in fairness, the authors acknowledge them in one way or another. As such, these are not flaws per se, but gaps that can be filled with further research.Experimental manipulations, such as controlled playback experiments or controlled environments, could strengthen the causal claims by directly testing the effects of specific maternal behaviors on vocal development. Certainly, the strengths of the paper will be consolidated after such work is performed.The reliance on the number of singing males as a proxy for social acoustic input. This measure does not account for the variability in the quality, frequency, or duration of the male songs to which the pups are exposed. A more detailed analysis of the acoustic environment, including direct measurements of song exposure and its impact on vocal learning, would provide a clearer understanding of the role of male tutors.Finally, and although it would be unlikely that these results are unique to Saccopteryx bilineata, the study's focus on a single species limits at present the generalizability of some of its findings to other vocal learning mammals. While the parallels drawn between bat and human vocal development are intriguing, the conclusions will be more robust when supported by comparative studies involving multiple species of vocal learners. This will help to identify whether the observed maternal influences on vocal development reported here are unique to Saccopteryx bilineata or represent a broader phenomenon in chiropteran, mammalian, or general vocal learning. Expanding the scope of research to include a wider range of species and incorporating cross-species comparisons will significantly enhance the contribution of this study to the field of vocal learning.

Thank you for your suggestions and comments.

Regarding your main comment 1: In the future, we plan to implement temporary captivity experiments to investigate how maternal behaviours affect pup vocal development. This study provides the necessary basis for conducting future playback studies investigating specific behaviours in a controlled environment.

Regarding your main comment 2: We completely agree that the number of singing males only represents a proxy for acoustic input that pups receive during ontogeny. In the future, we plan to investigate in detail how the acoustic landscape influences pup vocal development and learning. This will include quantifying how long pups are exposed to song during ontogeny and assessing the influence of different tutors, including a detailed analysis of song syllables of the adult tutors to compare it to vocal trajectories of song syllables in pups.

Regarding your main comment 3: We also fully agree that it is unlikely that these results are unique to *Saccopteryx bilineata.* We are certain that other mammalian vocal learners show parallels to the vocal development and learning processes of *S. bilineata*. Especially bats are a promising taxon for comparative studies because their vocal production and perception systems are highly sophisticated (due to their ability to echolocate). The high sociability of this taxon also includes a variety of social systems and vocal capacities (e.g. regarding vocal repertoire size, vocal learning capacities, information content, etc.) which support social learning and social feedback – as shown in our study.

As suggested, in our revised manuscript we have includes information on the validation of the ethogram. Furthermore, we have corrected all the spelling mistakes – thank you very much for pointing them out!

**Recommendations for the authors:**

**Reviewer #1 (Recommendations for the authors):**
The following comments and suggestions are offered to improve clarity and strengthen support for the paper's main claims.(1) Female displays as feedback:a) The authors rather broadly describe maternal behavior as feedback based on its occurrence during pup babbling. Feedback typically entails some degree of response contingency, which is not explicitly established here. Although the authors qualitatively describe a variety of female displays that only occur within the babbling context, they also state that "all these behaviors could occur singly or in an interactive way" (Line 102). The authors go on to use aggregate counts of these diverse female displays in their analyses. It would of course be interesting to know whether distinct female displays are evoked differentially by pup behavior and whether specific female behaviors, in turn, predict subsequent pup vocalizations. A display-specific approach might also reveal more about the mechanisms by which the female behavior shapes babbling (e.g. specific reinforcement signals vs. more graded social facilitation or 'audience effect'). However, even without identifying such finegrained contingencies, the main text should at least mention the results shown in Figure 1A. Namely, that pups initiate ~80% of interactive behavioral sequences, suggesting that subsequent maternal displays are likely to be pup-contingent responses (i.e. feedback) and not simply co-occurring behavior.

We fully agree with Reviewer 1 that it would be very informative to investigate whether distinct female displays are evoked differentially by pup behavior, such as specific syllables within babbling. Or conversely, whether specific female behaviors precede particular pup vocalizations. For this study, we documented maternal behavior through direct observations, field notes, and/or video recordings. However, to capture potential contingencies between specific maternal behavioral displays and vocalization occurring in the millisecond range, other data collection methods (e.g. high-speed camera) will be required in the future.

Related to this, we have included the following statements (see below). Statement 1 also cites a very recent study in zebra finches, demonstrating that female calls can promote song learning success (Bistere et al. 2024, line 57, lines 304-305).

Lines 297-305: This finding serves as an initial indication that non-vocal interactions with the mother may influence a pup´s individual learning trajectories. Future studies will focus on the relationship between acoustic change, maternal feedback, and learning success, specifically investigating contingencies between particular pup vocalizations and maternal displays in natural settings. Playback experiments are an additional approach to test the impact of contingency on vocal learning. For example, one study in zebra finches demonstrated that contingent non-vocal maternal feedback affects imitation success (Carouso-Peck & Goldstein, 2019), while another recent study found that female calls can promote song learning but the role of contingency remains to be determined (Bistere et al., 2024).

Lines: 332-334: This might also apply to *S. bilineata* where pups initiated ~ 80% of social interactions, suggesting that maternal feedback is likely influenced by the pup´s vocal practice.

b) The authors claim that the number of maternal displays during babbling predicts the duration of babbling bouts (Figure 1D). I find this analysis - and others based on the raw number of behaviors during babbling - difficult to interpret given that the raw number of displays may depend upon the duration of the babbling bout over which they are counted. In other words, might the number of displays reflect the fact that more displays can occur within the interval of longer babbling bouts? It would be relatively straightforward to minimize this potential confound by testing whether female display *rates* predict longer bouts.

We calculated the display rates (maternal displays per bout duration) and conducted a GLMM (the same analysis after log-transformation and scaling) like in our original manuscript (model 1).

GLMM

summary(vocpracf)

Generalized linear mixed model fit by maximum likelihood (Laplace Approximation) ['glmerMod'] Family: Gamma (log)

Formula: bout_dur ~ age.z + behavioural_quotient.log.z + nomales.z + (1 | ID) Data: set1

**Author response table 1. sa2table1:** 

AIC	BIC	loglik	deviance	df.resid
2594.9	2614.2	-1291.4	2582.9	180

**Author response table 2. sa2table2:** Scaled residuals.

Min	1Q	Median	3Q	Max
-1.4604	-0.7379	-0.1924	0.5238	2.6564

**Author response table 3. sa2table3:** Random effects.

Groups Name	Variance	Std.Dev.
ID (Intercept)	0.0569	0.2385
Residual	0.3664	0.6053

Number of obs: 186, groups: ID, 19.

**Author response table 4. sa2table4:** Fixed effects.

	Estimate	Std. Error	t value	Pr(> |z|)
(Intercept)	6.13150	0.09114	67.278	< 2e-16 ***
age.z	0.41835	0.05048	8.287	< 2e-16 ***
behavioural_quotient.log.z	-0.14627	0.05310	-2.754	0.00588 **
nomales.z	-0.06251	0.09473	-0.660	0.50934

Signif. codes: 0 ‘***’ 0.001 ‘**’ 0.01 ‘*’ 0.05 ‘.’ 0.1 ‘ ’ 1.

**Author response table 5. sa2table5:** Correlation of fixed effects.

	(Intr)	age.z	bh_.2.
age.z	0.010		
bhvrl_qt.2.	0.011	-0.318	
nomales.z	0.110	-0.110	0.001

Interpretation: Our analysis in the original manuscript shows that the bout duration increases with number of maternal displays. As reviewer 1 points out: more time offers more opportunities for the mother to show displays. The number of displays in longer bouts could just reflect that more displays are possible in a longer period. This could be a potential confounding factor. However, our analysis of display rates as an explaining factor shows that the relationship between bout duration and display rate is negative. This means that in longer bouts the displays increase (as seen in the first scenario), but they happen less frequently per time unit. This could indicate that in longer bouts, the mother takes breaks or longer periods of time between each display, which decreases the frequency of displays. This minimizes the risk of a potential confound, as it shows that the rate of displays tends to decrease rather than increase in longer bouts. In summary: The display rate does not appear to ‘favour’ longer bouts, as longer bouts are associated with a lower display rate. This speaks against the hypothesis that the number of displays only increases due to the longer bout duration. This also means that our analyses, which show that maternal displays influence song syllable production, are not biased or confounded by the bout duration. This suggests that maternal behaviour is targeted and selective, and represents a potentially contingent reaction to the pup´s vocal production, and is not simply determined by the duration of a bout.

We added this analysis in our supplementary material (Table S2) and pointed this out in the revision of our main manuscript (lines 136-138).

c) The introduction states that "Pup babbling is not tied to a specific function." (Lines 75-78). This may be an important point worth exploring with this unique data set. For example, the termination of a babbling bout is defined in some cases by the onset of nursing. Have the authors (or others) tested whether babbling elicits nursing behavior? If so, this may represent a reinforcement mechanism that affects babbling rates and subsequent song outcomes. Similar functional shifts in developing vocal behavior have been reported in male chipping sparrows, in which juvenile begging calls - which initially elicit parental feeding behavior - can later be incorporated into 'sub-song' (i.e. babbling) during the development of courtship song (Lui, Wada, Nottebohm, PLOS ONE, 2009).

Thank you for pointing out this interesting study on chipping sparrows!

To address your question: Strauss et al. (2010) conducted a study on pup and maternal behaviors, demonstrating that babbling did not consistently result in nursing. When denied care, pups often returned to resting or grooming, a pattern we also observed in our study. While nursing might provide an additional reinforcement mechanisms, it is not the cause that evokes babbling – this is what we mean by stating “pup babbling is not tied to a specific function”. Babbling is not a begging behavior as described by Lui et al. 2009. As mentioned in the review of ter Haar et al. 2021, babbling differs structurally from begging in that it is composed of both adult-like and juvenile syllables and lacks context specificity. To solicit care (i.e. begging) pups produce several isolation calls in a fast repetitive manner. We added a more detailed explanation to make this distinction clear (lines 79-83).

Another interesting fact and probably more comparable to the study of the chipping sparrows – in which begging calls are incorporated into subsong practice – might be the isolation call syllables of *S. bilineata*. Directly after birth, *S. bilineata* pups produce multisyllabic isolation calls (see Knörnschild & von Helversen 2008, Knörnschild et al. 2012, Fernandez & Knörnschild 2017) that serve to solicit maternal care. For the first 2.5 weeks, pups only produce innate vocalizations, including echolocation and isolation calls (Fernandez et al. 2021). During the babbling phase, the syllables encoding the individual (and group) signature of the isolation call are also incorporated into babbling bouts. The production of isolation calls might also mark an initial step in the vocal learning process. However, in contrast to the subsong of chipping sparrows, babbling bouts in *S. bilineata* also include syllables acquired through vocal imitation. Thus, although we find similarities in vocal practice and development between chipping sparrows and *S. bilineata,* there are also distinct differences.

(2) Are pups exposed to more male songs when the mother is present?The number of singing males in each colony was used as a reasonable proxy for the amount of social acoustic input. However, I wonder if pups are exposed to more adult male songs when the mother is present and, relatedly, if females tend to remain present for longer if a pup is babbling (potentially increasing its exposure to male songs during the babbling phase).

The mother is always present when males are singing. In *S. bilineata,* males predominantly engage in territorial song twice daily: at dusk and dawn. After foraging at night, territorial singing males are the first to return to the roost, and females will only return when they hear male song. Pups are either attached to the mother´s belly or – when growing older – will fly into the roost followed by the mother. In the evening, males sing approximately half an hour before leaving for foraging. Females will usually leave first, followed by their pups, and males leave last. Hence, females/mothers are always present when pups are exposed to male acoustic input.

(3) Pup sex differences:The authors test for sex differences within a subset of pups and briefly mention that vocal development is considered in both males and females. This presumably means that female pups also exhibit vocal imitation of adult male territorial songs, even though they only produce these vocalizations during the babbling phase, after which they stop singing entirely. If so, this would, to my knowledge, be a unique phenomenon among vocal learners and would be interesting to discuss in greater detail.

We followed your recommendation and discussed this topic in greater detail. We included the following part in our discussion (lines 257-269): An intriguing aspect of this species is that, unlike most song-learning songbird species, female pups show no differences from males in babbling behavior and vocal development (Fernandez et al. 2021). This study corroborated this finding: female pups received the same maternal feedback, and their song syllable imitation did not differ in any way from male pups (as observed as well in Knörnschild et al. 2010). This phenomenon is rare among vocal learners and raises the question of why female pups match male vocal development despite not using the learned vocalizations later in life. One potential explanation might lie in the function of the territorial song for adult females: it serves as an acoustic signal to help females locate new suitable colonies after dispersal. The territorial song exhibits different dialects, with females showing a preference for local over foreign dialects (Knörnschild et al., 2017). The own early practice and production of song might enhance the ability to evaluate male song and support mating decisions.

(4) Characterization of song syllables:The authors explain their acoustic analyses in detail within the methods, however, descriptions of the syllable classification procedures and acoustic movement analyses need to be presented more clearly in the main text, so readers unfamiliar with bioacoustics or previous work can follow the logic. Also, given the qualitative descriptions of the data and the two spectrogram examples provided (Figures 2 and S1), it is difficult for the reader to fully evaluate the suitability and output of these critical procedures.Suggestions:- Qualitative descriptions of syllable characteristics (i.e. buzz, pulse, trill, ripple, gap, smeared noisy, precursor syllable, mature syllable, adult-like syllable, early vs. late babbling phase, syllable name, etc) should all be clearly-labeled in example spectrograms and used consistently, without using different terms interchangeably (e.g. mature vs. adult-like).

We understand that we should provide a clearer description of the various terms essential to understanding this study. We added a “Terminology” box (line 158) to the main manuscript, defining the acoustic terms we are using throughout our study. Additionally, we enhanced Figure S1 by providing more detailed information on the spectrogram that displays the five distinct song syllable types. Moreover, we included an additional spectrogram in the supplementary material (Fig. S2) displaying examples of precursor and mature syllables for syllable B2. In the method section, “The acoustic movement during ontogeny”, we added a sentence clarifying the terms “early” and “late babbling phase” (Lines 605-606).

- Show as you tell. Plot the data, at least from a representative pup, for each major step in the analyses (labeled spectrogram, PCA plots with distinct syllable clusters, high vs. low versatility, precursor vs. mature variants, early vs. late syllables with Euclidean distances between centroids and relation to "generic" adult male syllables, etc.)

To illustrate the acoustic analysis more comprehensively, we have made the following additions:

-we included a Figure (Fig. S3) in the supplementary materials showing an excerpt of a babbling bout with labelled syllables to illustrate how we analyzed (a) total song syllable count per bout, (b) versatility per bout, and (c) the number of precursor versus mature B2 syllables (the most common syllable type).

-Additionally, we included a spectrogram with three exemplary B2 syllables to illustrate the acoustic parameter extraction with Avisoft SASLab Pro software for subsequent analysis of vocal change during development (Fig. S4 A).

Lastly, we included a DFA for one of the colonies with three exemplary pups to illustrate how we calculated each pup's acoustic change during ontogeny (Fig. S4 B).

(5) Minor Comments and Corrections:- Modeled data are log-transformed, however, the raw data are plotted on linear scales, and in most cases, data points are densely clustered and overlapping at lower values. Plotting the data on log scales would likely aid visibility.

We appreciate this suggestion and changed the plots accordingly.

- Figure 1E displays 18 data points, (legend says n=19).

The legend is correct; the figure includes 19 data points. Two mothers have the same activity score, so their points are at the same location and it looks like there are only 18 data points.

- Line 482: Is "VCL" media player meant to refer to "VLC" player?

Yes, thank you for spotting that. We corrected it.

**Reviewer #2 (Recommendations for the authors):**
I have only a couple of comments:- Perhaps it would be useful to briefly go over the validation used for the ethogram in Table S1.

The behaviors listed in the ethogram were defined based on Strauss et al. (2010) and expanded based on our own observations. For consistency, we developed these definitions and trained the students analyzing behavioral data for this study. During the training phase, we validated their analyses until the inter-observer-reliability reached 100% (lines 507-508).

- The paper seems to be generally written in American English, yet there are some instances of British English spelling, e.g. "standardised"/"standardisation": table 1, table 2, lines 143, 228, 524, 525, 531, 546, 547, 554, 560, 561.

Thank you for spotting these errors, we corrected them.

- Line 343: "at libitum" should be "ad libitum".

Thank you for spotting this error. We corrected it.